# Visual Feedback and Postural Control in Multiple Sclerosis

**DOI:** 10.3390/jcm9051291

**Published:** 2020-04-30

**Authors:** Hernan Inojosa, Dirk Schriefer, Katrin Trentzsch, Antonia Klöditz, Tjalf Ziemssen

**Affiliations:** MS Center, Department of Neurology, Carl Gustav Carus University Hospital, Technical University of Dresden, Fetscherstr. 74, 01307 Dresden, Germany; hernan.inojosa@uniklinikum-dresden.de (H.I.); Dirk.Schriefer@uniklinikum-dresden.de (D.S.); Katrin.Trentzsch@uniklinikum-dresden.de (K.T.); Antonia.Kloeditz@uniklinikum-dresden.de (A.K.)

**Keywords:** Multiple sclerosis, balance, postural control, visual feedback, static posturography, somatosensory impairment

## Abstract

As people with multiple sclerosis (pwMS) manifest heterogeneous demyelinating lesions that could affect somatosensory or vestibular ways, visual stimulus as feedback could be especially relevant to achieve postural control. This has clinical importance for the development of preventive measures and rehabilitation therapies in order to avoid falls and accidents in this group. In our study, we objectively evaluated the influence of visual feedback on the stabilization of balance in pwMS versus healthy controls (HC) and its potential utility in clinical evaluation. Static posturography tests were performed in 99 pwMS and 30 HC. Subjects stood on a force platform with open and closed eyes. During this procedure, three balance parameters were obtained for both vision conditions: average sway, average speed, and average speed of sway. Neurostatus-Expanded Disease Disability Score (EDSS) and Multiple Sclerosis Functional Composite (MSFC) were performed in parallel as well. A two-way mixed repeated measures ANCOVA, controlling for sex and age, was performed to evaluate the effect of vision, MS diagnosis, and the interaction of both in static posturography parameters. The difference between both closed and open eyes conditions was calculated for each parameter and further analyzed according to MS-relevant clinical variables. The magnitude of the vision effect differed between pwMS and HC as a significant interaction between the vision and the MS diagnosis in the delineated area (*p* < 0.001) and average speed of sway (*p* = 0.001) was seen. These parameters had a greater increase in pwMS than in HC after closing eyes. For the average sway, a significant main effect of vision was present (*p* = 0.047). Additionally, the differences obtained between open and closed eyes conditions assessed with the delineated area and average speed of sway were moderately correlated to the assessed clinical tests EDSS (*r* = 0.405 and *r* = 0.329, respectively) and the MSFC (*r* = −0.385 and *r* = −0.259, respectively). In our study, pwMS were more dependent of visual feedback than HC to maintain postural control. This easy and short evaluation by static posturography could support the development of targeted preventive measures and interventions in pwMS.

## 1. Introduction

Postural control is an essential neurological function for the execution of daily activities. It can be defined as the act of maintaining, achieving, or restoring a state of balance [1]. It is a complex ability that depends of a non-linear integration of different neurological functions including postural stretch reflexes, motor skills and sensory inputs, such as proprioception, vestibular, and visual afferences [2]. Function loss in one of these systems could alter the capacity of maintaining postural control, increasing the patient’s risk of falling with further social and economic impact [3,4].

Such a damage is observed especially among people with multi-systemic neurological diseases. A good example is multiple sclerosis (MS), a chronic inflammatory disease characterized by heterogeneity of symptoms and pathophysiological mechanisms [5,6]. Balance deficits and ambulatory impairment are key symptoms of the disease [7]. Inflammatory lesions can affect every function system and domain, and negatively affect postural control [5,8,9,10,11]. Lesions on somatosensory ways alter postural stability as this fundamental feedback is impaired, making a postural compensation and the visual feedback often more necessary to maintain balance [12,13,14,15].

The influence of the visual stabilization in the maintenance of posture is classically assessed with the Romberg test, a neurological examination used to evaluate balance in clinical practice [16]. It returns a positive result when the patient falls after closing eyes, suggesting an impairment on the proprioceptive way. Previous reports suggest that people with MS (pwMS), possibly with neuronal transmission impairment or lesions on somatosensory ways, are more dependent of this visual compensation for postural control [12,13,17,18], presenting thus a higher degree of imbalance and/or positive Romberg test results after closing eyes [11,19,20]. Reduced white matter of the cortical proprioceptive tracts [11], slower spinal somatosensory conduction [12], or supratentorial lesion volumes may ease imbalance in pwMS [9].

Static posturography is well accepted as an objective technique to assess balance function among pwMS and to quantify possible impairment. Balance parameters obtained with force platforms can differentiate between pwMS and healthy controls (HC), even in pwMS with minimal disability detected by neurologists [21,22,23,24,25]. However, the influence of visual compensation in postural balance in pwMS is poorly investigated by static posturography. A research group has already reported more frequent balance impairments in closed eyes condition in a MS population compared to HC [26]. A more recent study reported a greater reliance in proprioception and the visual system in pwMS than in HC [19]. The visual feedback in balance assessed with static posturography could then additionally offer different conditions and techniques for the generation of balance parameters as clinical outcomes to be used in clinical practice. Proper assessment of postural control in pwMS is therefore necessary to address optimized physiotherapeutic therapies and preventive measures to avoid falls due to imbalance. The confirmation of a predominant role of visual inputs in certain MS patients could help targeting these actions.

Our study aimed primarily to evaluate the influence of visual stabilization in pwMS using static posturography. In addition, we assessed the correlation of this visual feedback with clinical disability and function outcomes used in medical practice such as the Neurostatus-derived Expanded Disease Disability Score (EDSS) [27] and Multiple Sclerosis Functional Composite (MSFC) [28]. We hypothesized that pwMS would present a greater influence of visual feedback than HC, reflected as a higher increase in the evaluated balance parameters after assuming closed eyes condition. In addition, pwMS with somatosensory or cerebellar impairments were expected to be more dependent on this feedback than those without it. This analysis aims to support the evaluation of pwMS and the indication of balance improving measurements on behalf of a direct practical benefit for the patients.

## 2. Materials and Methods

We conducted a cross-sectional study in the Multiple Sclerosis Center at the Center of Clinical Neuroscience of the Department of Neurology, University Hospital Carl Gustav Carus, Dresden, Germany [25]. PwMS and healthy controls (HC) without neurological disease were invited to participate. Inclusion criteria were: (1) confirmed multiple sclerosis according to McDonald’s criteria; (2) EDSS Score between 0 and 5.0; (3) age between 18 and 50 years; (4) no acute attacks or cortisone treatment in the preceding three months period; (5) no congenital or traumatic visual impairment; and (6) no orthopedic disease of the lower limbs. PwMS with a diagnosis of relapsing remitting MS (RRMS) and secondary progressive MS (SPMS) participated in this study, as well as a group in transition from RRMS to SPMS (defined as RRMS with incomplete relapse recovery but unfulfilled criteria of SPMS). Written informed consent was provided. Each participant was examined according to good clinical practice (GCP) guidelines. The ethical committee of the Technical University of Dresden, Germany approved the procedures of this study (approval code: EK 224062011).

Neurostatus-certified physicians calculated EDSS step scores for every patient, including MS-specific visual, cerebellar, and sensory function systems [27]. As part of the Neurostatus-EDSS, temporary and permanent signs and symptoms that were not occasioned by MS were not taken into consideration when assessing the scores. PwMS were classified according to function systems (FS) as follows:PwMS with visual impairment: no visual impairment if visual function system was 0; with visual impairment if the visual function system was ≥1.PwMS with cerebellar impairment: no cerebellar impairment if cerebellar function system was 0; with cerebellar impairment if the cerebellar function system was ≥1.PwMS with sensory impairment: no sensory impairment if sensory function system was 0; with visual impairment if sensory function system was ≥1.

All subjects additionally completed a MSFC test, which included timed tests of ambulation (Timed 25-Foot Walking Test), arm function (Nine-Hole Peg Test), and cognition (Paced Auditory Serial Test) [28]. A summary z-score was generated including the performed tests.

Static posturography was performed using a commercially distributed force platform (Force Platform GK-1000, MediBalance Pro Test- and Trainingssystem, MediTECH Electronic GmbH, Wedemark, Germany). Four piezoelectric sensors in the platform transformed pressure impulses generated by the subject’s center of gravity in electrical impulses. Balance parameters were automatically generated by a software package. Each measurement had a total duration of 30 s and started after an adjustment period of 20 s standing on the electronic platform. Subjects stood barefoot on a marked area (modified with a track width of 10 cm) of the measurement platform as quiet as possible and extended their arms in front of them with palms facing up. Two conditions were assessed as patients adopted postures with open eyes and closed eyes conditions. A pause of 60 s between each assessment was made and each test was performed in a single measurement. The absolute and relative difference between both conditions was calculated for each balance outcome to assess the influence of visual feedback on postural stability.

The following balance parameters were obtained for both conditions:Delineated area: total described surface during the measurement of the center of gravity of the subject calculated with 95% confidence interval (measured in cm^2^).Average sway: average distance or fluctuation from the center of all measurements (in mm).Average speed: average speed at which the central pressure point of the subjects moves on the platform (measured in mm/s).

### Statistical Analysis

We assessed normality of data using quantile–quantile plots and confirmed it with Shapiro–Wilk tests. Balance variables were log transformed before analyses to stabilize variance and to optimize normality for (slightly) right-skewed distributions of balance outcomes. Quantitative population characteristics are presented as measures of central tendency (mean, median), followed by standard deviation (SD) or interquartile range (IQR). Categorical characteristics are expressed as relative frequencies. A descriptive specification of (crude) mean values and standard deviations occurred in the evaluation of balance parameters. Differences between both assessed conditions were calculated and presented as absolute values and relative (%) increase. To evaluate the effect of vision in postural control, a general linear mixed model (two-way mixed repeated measures ANCOVA), adjusted for sex and age, was performed assessing differences between pwMS and HC, as well as differences within each group in the balance performance in open and closed eyes conditions. The defined factors included in this analysis were “vision” (within-subject measure defined by the performances for each balance parameter in the two conditions: open and closed eyes) and “MS diagnosis” (between-subjects measure, classifying subjects in two groups: pwMS and HC). Necessary assumptions for performing the selected models were checked, such that a linear relationship between covariables and the dependent variable (balance outcomes) was confirmed using scatterplots. Homogeneity of regression slopes was confirmed before conducting the analysis. Mauchly’s test of sphericity did not indicate a violation of the assumption of sphericity as only two levels of the within-subject factor (vision conditions) occurred. Having identified main effects for the assessed factors and their interaction, Bonferroni-based post-hoc comparisons were carried in order to determine simple (main) effects of the interaction term. Simple (main) effect analyses indicate differences in one factor (vision) at each level of another factor (MS diagnosis) and vice versa. Corresponding adjusted *p*-values and effect sizes were reported. In all models, partial eta-squared (ηp^2^) were interpreted as a measure of effect size as small (ηp^2^ > 0.01), medium (ηp^2^ > 0.06), or large (ηp^2^ > 0.14) [29,30]. Spearman rank correlations were calculated to study bivariate relations of balance outcomes with EDSS, visual, cerebellar, and sensory FS and with MSFC scores. Additionally, unpaired *t*-tests were performed including only pwMS with or without disability in EDSS-derived FS. Cohen’s *d* effect sizes were additionally calculated and interpreted as small (*d* = 0.2), medium (*d* = 0.5), or large (*d* = 0.8) [29,30]. Significant results were those with significance levels of *p* < 0.05. All statistical analyses were performed using IBM SPSS version 25.0 (IBM Corporation, Armonk, NY, USA). Graphics were generated with GraphPad Prism 5 (GraphPad Software, Inc., La Jolla, CA, USA).

## 3. Results

A group of 129 subjects participated in this study: 99 pwMS and 30 HC. The mean age of the pwMS group was 35.01 years (SD 8.21), 68.7% were female, with a median EDSS of 2.0 in a range between 1.0 and 5.0, cerebellar function system of 0 (between 0 and 3), and sensory function system of 1 (range between 0 and 3). PwMS and HC did not differ based on age or gender (Table 1). The healthy group had a better performance in the MSFC as expected (*p* < 0.001).

### 3.1. Postural Stability According to Visual Stimulus in People with MS and Healthy Controls

The results of the two-way mixed ANCOVA showed that there was a significant main effect of the visual feedback in postural control, but it differed between pwMS and the HC (Table 2 and Table 3). There was a significant interaction between the vision and MS diagnosis (vision*MS diagnosis) observed with the delineated area (*p* < 0.001) and average speed of sway (*p* = 0.001), but not in the average sway (*p* = 0.090). In the average sway, no significant interaction was observed, but the main effect of vision showed a statistically significant difference between the groups (*p* = 0.047).

Figure 1 shows Bonferroni adjusted simple (main) effects analyses for the vision (open, closed eyes) * MS diagnosis (PwMS, HC) interactions for the delineated area and average speed of sway, as well as for the vision main effect for the average sway as post-hoc tests to the two-way mixed repeated measures ANCOVA results (Table 3). Differences between PwMS and HC in each assessed condition and differences within subjects for each balance parameter are seen. Considering the delineated area, significant differences were seen between PwMS and HC in both open and closed eyes conditions, with higher effect sizes in the closed eyes condition (Figure 1a). In addition, simple effects of vision were significant within PwMS and HC, with higher effect sizes in the MS group (Figure 1a). A similar pattern was seen for the average speed of sway, where PwMS and HC showed significant differences in both vision conditions and the vision main effect differed within the groups, with larger effect sizes in this latter for PwMS (Figure 1c). Considering the average sway, although no significant interaction was seen, there was a significant main effect of vision. A post-hoc analysis showed significant differences between the open and closed eyes performance for the PwMS group (*p* < 0.001) and not for the HC (*p* = 0.871; Figure 1b).

### 3.2. Correlation of Balance Outcomes According to Visual Conditions with Established MS Disability and Function Tests

The evaluated balance parameters and their alterations after closing eyes were analyzed in context of the EDSS (including function systems calculated as part of it) and the MSFC (Table 4). The delineated area had the highest correlations with the evaluated tests overall, with significant correlations with the EDSS and MSFC in open and closed eyes conditions, as well as for the difference after closing eyes. Similarly, the average speed of sway had significant correlations with the clinical tests for every assessed performance. Differently, even though the average sway had significant correlations with the EDSS and MSFC in open and closed eyes conditions, the differences between both vision conditions had no significant correlations with the rest of the assessed clinical outcomes. The differences obtained in the delineated area and average speed of sway after closing eyes had significant correlations with all clinical tests with exception of the visual FS. However, for the average sway, the difference between vision conditions did not correlate with any of the evaluated outcomes (Table 4).

Additionally, the correlation between the delineated area and average speed of sway (*r* = 0.802, *p* < 0.001) was very high. Conversely, the correlation was lower between these parameters and the average sway (*r* = 0.318, *p* = 0.001, and *r* = 0.320, *p* = 0.001, respectively).

The MSFC was significantly and inversely correlated with all three parameters in open and closed eyes conditions, with higher correlations with the delineated area with closed eyes (*r* = −0.422) (Table 4).

### 3.3. Influence of Visual Impairment in Postural Control According to EDSS Function Systems

PwMS were additionally classified according to the presence of visual, cerebellar, or sensory impairments evaluated with the EDSS. Patients with sensory impairment assessed by the neurologist had a higher increase in their delineated area (*p* = 0.004), average sway (*p* = 0.011), and average speed of sway (*p* = 0.026) than those without it (Table 5). Regarding the other function system impairments, only the difference in the average speed of sway was significantly higher in pwMS with cerebellar function impairment than in those without impairment in this system (*p* = 0.031).

## 4. Discussion and Conclusions

In our study, we demonstrated a relationship between the visual feedback and postural control in pwMS and in healthy controls using objective outcomes obtained with static posturography. Both groups had higher balance parameters standing with closed eyes than with open eyes. However, the visual stimulus was more relevant on postural control in pwMS than in HC, with a significant interaction between vision and the MS diagnosis for two of the evaluated balance outcomes as assessed with a two-way mixed repeated measures ANCOVA.

Our results are in line with previous reports, which suggest greater instability in pwMS compared to healthy controls [18,19,25,26,31]. Both groups were more unstable in the closed eyes condition in all balance parameters. Nonetheless, a significant interaction between vision and the MS diagnosis for the delineated area and average speed of sway after closing eyes was observed, where simple main effect calculations evidenced higher effect sizes in the PwMS than in HC overall. For the average sway, no interaction was seen, but a significant main effect of vision was present. With these parameters, we observed that pwMS depend more on the visual stimulus for the maintenance of a stable posture than the healthy group. This could be related to the multiple pathophysiological mechanisms and heterogeneous lesions on different systems present in MS patients. Impairment of e.g., cognitive, motor, vestibular, or somatosensory functions, with slower axonal conduction, could make pwMS more dependent on visual inputs to achieve postural control [32,33]. Furthermore, previous studies assessing static posturography and somatosensory evoked potentials have reported that lesions, specially in proprioceptive ways (more than in cerebellum or pyramidal system), are mainly responsible for postural instability in pwMS [12,32,34]. An interesting study suggested that the instability between pwMS with opened or closed eyes may have further multiple anatomical pathomechanisms [26].

The somatosensory feedback is provided by group I (principally) and II muscle spindles and appears to be particularly degraded in pwMS and be the cause of imbalance in these patients [11]. In this order of ideas, we additionally evaluated the static posturography outcomes according to three function systems calculated by the neurologists in the EDSS examination. Corresponding with described pathophysiological mechanisms, visual stabilization was observed to be more relevant for pwMS with a sensory impairment than those without it. This is consistent with the classically performed Romberg test, where a positive result suggests impairment of the aforementioned somatosensory ways. We could demonstrate a relationship between the clinical evaluation of sensory function and the increase of the assessed balance outcomes.

Additionally, in the interpretation of the calculated effect sizes, we observed that the magnitude of the main effects of vision, the presence of MS diagnosis, and the interaction between both factors was medium for the delineated area and average speed of sway and low for the average sway. Interestingly, in the post-hoc effect analysis, we observed that balance parameters showed appreciably higher effect sizes in pwMS than in HC, especially for the delineated area and average speed of sway. A similar pattern was seen when evaluating only pwMS with or without disability in EDSS-derived sensory FS. The delineated area and average speed of sway could thus differentiate the analyzed groups with greater magnitude than the average sway. Moreover, the delineated area and average speed of sway showed higher correlations with clinical measures in pwMS, and the difference between the two vision conditions showed significant correlations only for these two balance outcomes; these results are consistent with the significant vision*group interaction obtained in the ANCOVA analysis. We consider that these differences could be associated to technical characteristics evaluated by these parameters, as their generation involves different methods and measures in the force platform, and previous studies suggest a better performance of outcomes involving area and velocity measurements [35,36]. Our results indicate that the delineated area and the average speed of sway may have the best performance in the assessment of pwMS. However, this was a cross-sectional observational study and the use of static posturography and the generation of balance parameters has been not yet standardized. To our knowledge, no previous study has reported effect sizes adopting a similar approach as our study to support the interpretation of our results.

These results have clinical relevance in the care of pwMS. Studies suggest that MS patients respond to impairment in somatosensory ways with certain compensation mechanisms, involving postural response with gastrocnemius contraction or ankle control [12,32]. Further interventions, including among others proprioceptive, motor, or endurance training, could be implemented to improve postural control in pwMS [37], especially in those patients with an impaired visual stabilization, making possible a targeted approach in gait interventions. Medications that impair postural control could be specially avoided in patients with greater somatosensory impairment and dependence on visual stimulation for postural control. Consequently, the assessment of the visual function in balance, together with other posturography parameters, could help as a tool in the prescription and therapeutic monitoring of these interventions.

Certain limitations should be considered in the interpretation of our results. Firstly, we were limited to balance outcomes obtained with a commercially distributed force platform. A supplementary evaluation with other static posturography outcomes could be performed to assess the influence of visual stimulation in pwMS. Second, there is no standard in the static posturography methods or instruments in MS. We adapted our trial to the routine care of pwMS in the MS Center Dresden, with a single measurement after a standard adaptation period. Repeated and posteriorly averaged trials are a valid alternative to be considered, and further studies could evaluate alternatives in the assessment of pwMS with the platform used. Third, although the EDSS is currently the standard clinical assessment tool for disability in pwMS and we included the MSFC in our study, further clinical measures could be used to evaluate the degree of somatosensory impairment, its correlation with the static posturography, and their differences after retiring the visual feedback. Furthermore, the EDSS evaluation does not include a specific ophthalmological assessment. We recommend considering this aspect in future studies. Fourth, due to the cross-sectional design of our study, a longitudinal evaluation was not performed, which could further estimate the practical and prognostic value of the assessment of visual feedback in pwMS and its long-term relevance in their treatment and follow-up. Finally, although we evaluated a group of pwMS with a low degree of disability, further analysis could be performed with patients with undetectable disease disability.

We encourage an objective assessment of postural control in pwMS with static posturography including both conditions and assessing the different between them as a possible clinical outcome marker. Further studies are needed to assess the utility of these outcomes in the long-term monitoring and therapeutic planning of MS treatment.

## Figures and Tables

**Figure 1 jcm-09-01291-f001:**
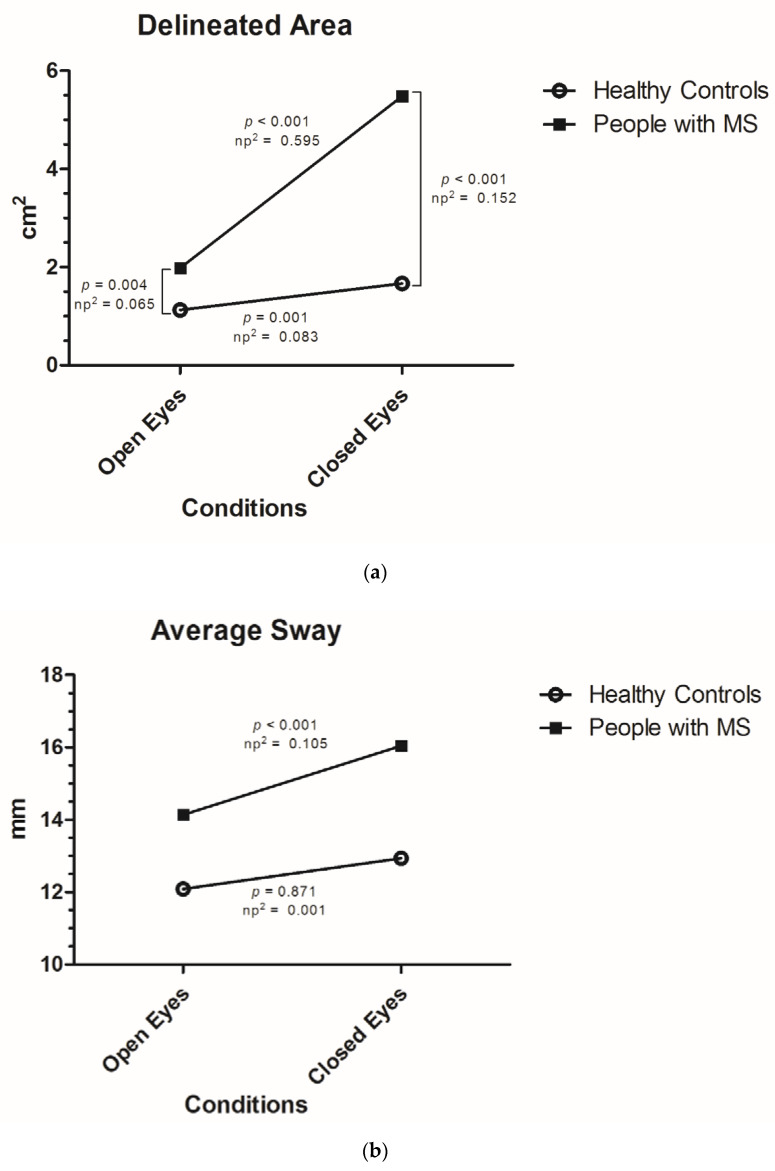
Graphical overview of the interaction effects and associated simple (main) effect analyses as Bonferroni-based post-hoc comparisons. Simple effect analyses revealed the degree to which one factor (vision) is differently effective at each level of another factor (multiple sclerosis (MS) diagnosis), and vice versa. In Figure (**a**) and (**c**), the interaction was significant and simple effects analyses revealed significant differences between people with MS (PwMS) and healthy controls (HC) in both conditions, as well as significant differences between the closed and open eyes performance for both groups. Higher effect sizes were seen for the vision effect in PwMS and MS diagnosis effect in closed eyes condition. In Figure (**b**), the interaction was not significant as visually indicated by almost parallel lines, but the main effect of vision was statistically significant. A simple (main) effect analyses was also carried out in (**b**) in order to assess the magnitude of the effect of vision in dependence of the level of MS diagnosis in more detail. See Table 2 for descriptive statistics in detail and Table 3 for exact *p*-values of the interaction effects. np^2^ = Partial Eta-squared.

**Table 1 jcm-09-01291-t001:** General characteristics.

	PwMS (*n* = 99)	HC (*n* = 30)	
Mean age in years	35.01 (SD 8.21)	34.03 (SD 7.98)	*p* = 0.892
Females	68 (68.69%)	21 (70%)	*p* = 0.563
Years Since Diagnosis (mean, SD)	5.48 (SD 4.62)	n.a.	
MS Subtype			
RRMS	91.9%	n.a.	
Transition to SPMS	7.1%	n.a.	
SPMS	1.0%	n.a.	
EDSS (median, IQR)	2.0 (IQR 1.50–3.0)	n.a.	
Visual Function System (median, IQR)	0 (IQR 0–1)	n.a.	
PwMS with visual impairment (*n*, %)	35 (35.3%)	n.a.	
Cerebellar Function System (median, IQR)	1 (IQR 0–1)	n.a.	
PwMS with cerebellar impairment (*n*, %)	57 (57.8%)	n.a.	
Sensory Function System (median, IQR)	1 (IQR 0–2)	n.a.	
PwMS with sensory impairment (*n*, %)	70 (70.7%)	n.a.	
MSFC Z-score (mean, SD)	0.602 (SD 0.421)	0.913 (SD 0.164)	*p* < 0.001

PwMS: people with multiple sclerosis. HC: healthy controls. MS: multiple sclerosis. RRMS: relapsing remitting multiple sclerosis. SPMS: secondary progressive multiple sclerosis. EDSS: expanded disease status scale. MSFC: multiple sclerosis functional composite. SD: standard deviation. IQR: interquartile range. n.a.: not available.

**Table 2 jcm-09-01291-t002:** Static posturography parameters in open and closed eyes conditions and difference between both in PwMS and HC.

Balance Outcome	Total (*n* = 129)	PwMS (*n* = 99)	HC (*n* = 30)
**Delineated Area (cm^2^)** (mean, SD)			
Open eyes	1.78 (SD 1.47)	1.98 (SD 1.61)	1.13 (SD 0.54)
Closed eyes	4.59 (SD 6.90)	5.48 (SD 7.65)	1.67 (SD 0.98)
Increase	+2.81 (SD 6.06); +158%	+3.50 (SD 6.76); +177%	+0.54 (SD 0.88); +48%
**Average Sway (mm)** (mean, SD)			
Open eyes	13.66 (SD 7.33)	14.14 (SD 7.85)	12.09 (SD 5.07)
Closed eyes	15.31 (SD 7.49)	16.04 (SD 7.59)	12.94 (SD 6.75)
Increase	+1.65 (SD 5.32); +12%	+1.90 (SD 5.62); +13%	+0.84 (SD 4.18); +7%
**Average Speed of Sway (mm/s)** (mean, SD)			
Open eyes	14.83 (SD 6.33)	15.49 (SD 7.00)	12.66 (SD 2.30)
Closed eyes	22.49 (SD 13.42)	24.40 (SD 14.66)	16.22 (SD 3.97)
Increase	+7.67 (SD 8.94); +52%	+8.91 (SD 9.77); +45%	+3.56 (SD 2.75); +28%

PwMS: People with multiple sclerosis. HC: Healthy Controls.

**Table 3 jcm-09-01291-t003:** Summary of two-way mixed ANCOVA results across static posturography balance parameters.

	Source	df	Mean Square	F	*p*	ŋp^2^
Delineated Area	Vision	1	0.454	14.102	<0.001	0.101
	MS Diagnosis	1	3.303	18.140	<0.001	0.127
	Vision * MS Diagnosis	1	0.415	12.890	<0.001	0.093
Average Sway	Vision	1	0.069	4.014	0.047	0.031
	MS Diagnosis	1	0.225	2.678	0.104	0.021
	Vision * MS Diagnosis	1	0.050	2.911	0.090	0.023
Average Speed of Sway	Vision	1	0.119	23.628	<0.001	0.159
	MS Diagnosis	1	0.451	10.892	0.001	0.080
	Vision * MS Diagnosis	1	0.062	12.277	0.001	0.089

Summary of 2 * 2 (Vision * MS Diagnosis) Mixed repeated measures ANCOVA results on performance in static posturography balance parameters. MS: multiple sclerosis.

**Table 4 jcm-09-01291-t004:** Correlation of static posturography outcomes with clinical stablished tests in PwMS.

Balance Parameter	EDSS	Visual Function System	Cerebellar Function System	Sensory Function System	MSFC
**Open Eyes**					
Delineated Area	0.327	0.113	0.262	0.287	−0.358
	*p* < 0.001	*p* = 0.269	*p* = 0.009	*p* = 0.004	*p* < 0.001
Average Sway	0.266	0.116	0.060	0.166	−0.342
	*p* = 0.008	*p* = 0.255	*p* = 0.555	*p* = 0.101	*p* < 0.001
Average Speed of Sway	0.285	0.207	0.275	0.299	−0.299
	*p* = 0.004	*p* = 0.041	*p* = 0.006	*p* = 0.022	*p* = 0.003
**Closed Eyes**					
Delineated Area	0.427	0.132	0.396	0.334	–0.422
	*p* < 0.001	*p* = 0.194	*p* < 0.001	*p* < 0.001	*p* < 0.001
Average Sway	0.330	0.130	0.160	0.286	−0.384
	*p* < 0.001	*p* = 0.200	*p* = 0.114	*p* = 0.004	*p* < 0.001
Average Speed of Sway	0.334	0.120	0.343	0.306	–0.293
	*p* < 0.001	*p* = 0.241	*p* = 0.001	*p* = 0.002	*p* = 0.003
**Difference**					
Delineated Area	0.405	0.091	0.379	0.361	−0.385
	*p* < 0.001	*p* = 0.375	*p* < 0.001	*p* < 0.001	*p* < 0.001
Average Sway	0.042	–0.034	0.118	0.090	0.0003
	*p* = 0.683	*p* = 0.741	*p* = 0.244	*p* = 0.375	*p* = 0.997
Average Speed of Sway	0.329	0.002	0.284	0.297	−0.259
	*p* < 0.001	*p* = 0.987	*p* = 0.004	*p* = 0.003	*p* = 0.010

EDSS: Expanded Disability Status Scale. MSFC: Multiple Sclerosis Functional Composite. Visual, cerebellar and sensory function system were calculated as part of the EDSS in the neurological examination.

**Table 5 jcm-09-01291-t005:** Influence of visual feedback in static posturography outcomes in pwMS according to EDSS function systems.

Difference after Withdrawal of Visual Stimulus	No Visual Impairment (*n* = 63)	With Visual Impairment (*n* = 35)	*p* (Effect Size)
Difference Delineated Area (cm^2^)	3.18 (SD 5.40)	4.06 (SD 8.82)	*p* = 0.963 (0.120)
Difference Average Sway (mm)	2.09 (SD 5.53)	1.53 (SD 5.91)	*p* = 0.649 (0.097)
Difference Average Speed of Sway (mm/s)	8.65 (7.67)	9.52 (SD 12.93	*p* = 0.876 (0.100)
	No cerebellar impairment (*n* = 42)	With cerebellar impairment (*n* = 57)	
Difference Delineated Area (cm^2^)	2.28 (SD 4.18)	4.40 (SD 8.07)	*p* = 0.057 (0.329)
Difference Average Sway (mm)	1.22 (SD 6.02)	2.40 (SD 5.30)	*p* = 0.307 (0.208)
Difference Average Speed of Sway (mm/s)	7.39 (SD 8.15)	10.04 (SD 10.75)	*p* = 0.031 (0.277)
	No Sensory impairment (*n* = 29)	With Sensory impairment (*n* = 70)	
Difference Delineated Area (cm^2^)	1.55 (SD 2.39)	4.31 (SD 7.76)	*p* = 0.004 (0.480)
Difference Average Sway (mm)	−0.01 (SD 3.98)	2.69 (SD 6.02)	*p* = 0.011 (0.851)
Difference Average Speed of Sway (mm/s)	6.32 (SD 7.41)	9.99 (SD 10.46)	*p* = 0.026 (0.404)

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
