# Peer review of "Visual Feedback and Postural Control in Multiple Sclerosis"

_jcm, 2020, doi:10.3390/jcm9051291_

Round 1

Reviewer 1 Report

Inojosa and colleagues reported on the effect of visual feedback on postural control in multiple sclerosis. The manuscript is overall clear and well written. Methods are sound. However, I have some issues I would like the authors comment on.

In the methods, perhaps, it is worth mentioning that visual, cerebellar and sensory function systems are specifically related to MS. In particular, there are a number of conditions which can affect visual function independently of MS (e.g., refractive errors, eye disease): how did authors exclude concomitant eye disease? Among inclusion criteria there is “no congenital or traumatic visual impairment”, which is rather general. Also, I wonder whether history of previous optic neuritis was collected.

From a statistical point of view, also considering the log-transformation of postulography variables, I would use ANOVA and linear regression models including age, sex and EDSS as covariates. In particular, EDSS should be accounted for, considering that the global motor function (e.g., pyramidal) can also affect balance. I do not think these analyses would actually change study results, but would definitely strengthen the main hypothesis of the paper.

In the text and figures, I would refer to open eyes and closed eyes, rather than condition 1 and condition 2.

In Table 1, it seems the low IQR for EDSS is 1.25, which does not correspond to an actual EDSS step. Could you please double-check?

In Figure 2 title, I would say that is “assuming”.

This is a cross-sectional study and, thus, speculations on causative effects should be avoided. In particular, I would remove words such as “influence” from the discussion.

Author Response

Dear Reviewer,

please see the attachment for our point-by-point response.

Thank you for your feedback.

Reviewer 2 Report

The current study investigated postural control function in pwMS with and without visual information. As compared with healthy individuals, pwMS group showed greater delineated area and speed of sway for both visual information conditions, these deficits increased in no vision condition. Although the findings seem to be very interesting, there are several issues that should be considered.

Major issues

Line 152: The statistical analyses (i.e., independent t-test and chi-squared test) on the first findings are redundant. Why did the authors perform multiple t-tests that may increase type 1 error? Two-way mixed ANOVAs (Group * Vision: 2 * 2) can be suitable for these findings. Based on this revision, the author may revise Figure 1.

Line 193: Why did the authors use different approaches between Table 2 (i.e., using dependent variables for each vision condition and Table 4 (i.e., using differences in dependent variables across vision conditions)?

Discussion: Please discuss any specificities of findings from three outcome measures (i.e., delineated area, average sway, and average speed of sway).

Minor issues

Line 113: Did the participants randomly perform the balance tests across the vision conditions?

Line 113: How many trials were administered for each condition?

Please replace “Condition 1 and 2” with “Open Eyes and Close Eyes” throughout the manuscript.

Table 3: Please add P-values for each correlation coefficient value.

Author Response

Dear Reviewer,

please see the attachment with our point-by-point response. We have addressed all you recommendations.

Thank you for your feedback.

Round 2

Reviewer 1 Report

Authors have addressed my concerns.

Author Response

Thank you again for your helpfull feedbacks.

Reviewer 2 Report

Although the authors addressed most of my concerns and extensively revised their manuscript, the remaining issues need to be carefully considered.

  1. For Table 2, the authors should state their findings based on the statistical results. For the Delineated Area and Average Speed of Sway, there were significant Vision*MS Diagnosis interactions. Please add specific methods for the post-hoc on these interactions, and state the findings based on the post-hoc analyses results. Further, there should be error bar (e.g., standard error) and symbols indicating statistical significant across vision conditions and groups in Figures 1A and 1C.
  2. The analysis on the Average Sway only revealed Vision main effects. Accordingly, Figure 1B should be revised with vision main effects findings (e.g., bar graph).
  3. Now, Table 3 seems to be redundant with Figure 1.
  4. Did you check the sphericity assumption for the ANCOVA findings?
  5. Given that the ANCOVA findings revealed significant vision and group interaction on the Delineated Area and Average Speed of Sway, the authors need to focus on the correlation findings regarding difference in vision conditions for the Delineated Area, Average Sway, and Average Speed of Sway (Part 3 of the Table 4). Interestingly, the Part 3 in correlation findings showed significant correlations for only the Delineated Area and Average Speed of Sway consistent with their ANCOVA findings.
  6. Table 6 seems to be statistically redundant with Table 5.
  7. The investigators chose to analyze a single trial from each individual within the groups.  Why only take one trial? The authors should justify this issue. 
  8. The abstract should be extensively revised based on their new statistical findings.
  9. In Table 2, MS should be stated with full name (mean square) because the author also use MS as a group.

Author Response

Dear Reviewer,

thank you very much for your helpful review report. We have edited our manuscript to address your commentaries. Please see the attachment for a point-by-point response.
